# Conformational Dynamics of Glucagon-like Peptide-2 with Different Electric Field

**DOI:** 10.3390/polym14132722

**Published:** 2022-07-03

**Authors:** Jingjie Su, Tingting Sun, Yan Wang, Yu Shen

**Affiliations:** Department of Applied Physics, Zhejiang University of Science and Technology, No. 318 Liuhe Road, Hangzhou 310018, China; sujingjie0617@163.com (J.S.); shenyu@zust.edu.cn (Y.S.)

**Keywords:** Glucagon-like peptide-2, molecular dynamics simulation, conformational change, electric field

## Abstract

Molecular dynamics (MD) simulation was used to study the influence of electric field on Glucagon-like Peptide-2 (GLP-2). Different electric field strengths (0 V/nm ≤ *E* ≤ 1 V/nm) were mainly carried out on GLP-2. The structural changes in GLP-2 were analyzed by the Root Mean Square Deviation (*RMSD*), Root Mean Square Fluctuation (*RMSF*), Radius of Gyration (*Rg*), Solvent Accessible Surface Area (*SASA*), Secondary Structure and the number of hydrogen bonds. The stable α—helix structure of GLP-2 was unwound and transformed into an unstable Turn and Coil structure since the stability of the GLP-2 protein structure was reduced under the electric field. Our results show that the degree of unwinding of the GLP-2 structure was not linearly related to the electric field intensity. *E* = 0.5 V/nm was a special point where the degree of unwinding of the GLP-2 structure reached the maximum at this electric field strength. Under a weak electric field, *E* < 0.5 V/nm, the secondary structure of GLP-2 becomes loose, and the entropy of the chain increases. When *E* reaches a certain value (*E* > 0.5 V/nm), the electric force of the charged residues reaches equilibrium, along the *z*-direction. Considering the confinement of moving along another direction, the residue is less free. Thus, entropy decreases and enthalpy increases, which enhance the interaction of adjacent residues. It is of benefit to recover hydrogen bonds in the middle region of the protein. These investigations, about the effect of an electric field on the structure of GLP-2, can provide some theoretical basis for the biological function of GLP-2 in vivo.

## 1. Introduction

Glucagon-like Peptide-2 (GLP-2) is a gastrointestinal hormone secreted by L cells in the gut and released into the bloodstream after the body eats food. It has a special effect on the intestinal development of newborn animals. GLP-2 is a 33-amino acid polypeptide corresponding to the position 126 to 158 of proglucagon, whose total structure weight is 3.77 kDa, containing 264 atoms, which is a single chain protein [1,2]. Crystal structure of GLP-2 has been obtained by nuclear magnetic resonance (NMR) spectroscopy [3]. The intermediate region of GLP-2 is a stable α-helix structure. However, the C-terminal region is loose [4], and experiments have shown that it can form a butt joint with the *N*-terminal of an extracellular receptor domain (Nt-ECO) with polymer, Nt-ECO belongs to the class B G-protein-coupled receptor family and plays a key role in glucose homeostasis and the pathophysiology of type 2 diabetes [5,6].

GLP-2 can inhibit the production of proinflammatory cytokines in the small intestine and serum of piglets lacking lipase. GLP-2 has an additive effect on EC proliferation, tissue growth, histomorphology and angiogenesis [7]. GLP-2 is involved in blood glucose homeostasis regulation through multiple pathways [7,8]. Many advantages of GLP-2 indicate that GLP-2 has a good clinical application prospect [9]. However, few studies attempt to investigate the dynamics of GLP-2.

Molecular dynamics have been widely used to explore the influence of external conditions such as temperature, ion, electric and magnetic fields on the structural stability of proteins in recent years [10,11,12]. According to Newtonian mechanics, the information of the structure, dynamics and thermodynamics of biological macromolecules can be obtained [13,14,15,16,17]. Several studies have demonstrated that MD simulations can replicate the dynamic aspects of some protein or peptides. [16,17].

As is well known, living systems can generate weak electric fields, so any cell membrane in its natural environment will also be constantly exposed to an electric field. Therefore, the electric field is forced to play an important role in the structure of proteins [18,19,20]. For instance, Jiang et. al. used molecular dynamics simulation to explore the influence of the applied electric field on the structure of 1BBL protein. It was found that *RMSD* and *Rg* of 1BBL increased as the external electric field increased. The protein was stretched under the action of the electric field, and the structure of 1BBL was obviously expanded [20]. Svejgaard et. al. explored the role of GABAB receptors in pro-epileptic and anti-epileptic effects of an applied electric field in rat hippocampus in vitro, and they have found in hippocampal slices that a single, very short (10 ms) electric field can suppress hyperactivity in a Cs+ model of epilepsy [21]. It was demonstrated for the first time that when the protein-DNA complex was exposed to an electric field (*E* < 1 kV/cm), the monomolecular disintegration rate constant of the complex increased, and resulted in the instability of the protein-DNA complex [22]. At the same time, there are also other investigations of protein-like chains under different electric fields [23]. Thus, focusing on the effect of electric field on the conformations of proteins is extremely significant. 

GLP-2 has important biological functions and possesses a beautiful helical structure. Although its biological functions have been studied extensively, there are few studies on its conformational changes and no relevant articles on molecular dynamics studies. Thus, it is necessary to investigate the effect of external physical conditions on the spatial structure of GLP-2. Therefore, it is quite significant to obtain the conformational changes in GLP-2 under different electric fields using molecular dynamics simulations.

## 2. MD Simulations and Methods

### 2.1. MD Simulation

Molecular dynamics simulations were applied to study the effect of external electric fields on GLP-2. The GLP-2 (PDB ID: 2l64) crystal structure can be obtained from the protein structure database (PDB) [1,2]. Uniform electric field was applied with the electric components *E* in the *z*-direction. Therefore, the modified equation of motion is given by
(1)$$m_{i}{\ddot{\vec{r}}}_{i}={\vec{f}}_{i}+q_{i}{\vec{E}}_{z}(t)$$

Here, $q_{i}$ denotes the charge of atom *i*, and ${\vec{f}}_{i}$ is the force on atom *i* due to the potential [24]. We performed molecular dynamics simulation on GLP-2 under seven different electric field conditions along *z*-direction: *E* = 0 V/nm, *E* = 0.2 V/nm, *E* = 0.4 V/nm, *E* = 0.5 V/nm, *E* = 0.6 V/nm, *E* = 0.8 V/nm and *E* = 1 V/nm. All MD simulations were performed using the GROMACS package (v.5.1.2, Royal Institute of Technology and Uppsala University, Stockholm and uppsala, Sweden) [25]. The Charmm36 force field and TIP3P water model were adopted for the protein and solvents, respectively. The simulation is under the NPT ensemble and atmospheric pressure [26,27,28]. First, the PDB conformation files were converted into GROMACS format structure files (.gro files), and the system topology files of GROMACS (.top) were generated. Protein was put into the box of the periodic cube, and TIP3P water molecules were added into the box to simulate the physiological environment of the protein. The minimum distance between the protein and the edge of the box was set as one nm. The long-range electrostatic interaction force was calculated by the PME method, and the list of non-bond interaction pairs was updated every 10 steps of the simulation. The width of the grid points was set as 0.16 nm, the L-J interaction was calculated by the cut-off method, and the truncation distance was set as 1.0 nm. In order to eliminate the unreasonable energy barrier generated by adding water molecules, the steepest descent method was applied to perform 50,000 steps to optimize energy. Four sodium ions were added to the system to keep it electrically neutral. Then, the seven systems were kept at a temperature of 300 K and a pressure of one bar, and the temperature and pressure were kept stable by weak coupling action. The coupling time of temperature and pressure was set at 0.1 ps, and the simulation step size was two fs. The molecular dynamics simulation of 50 million steps was carried out for the seven systems, which means the total simulation time is 100 ns. In this study, the Precision 3630 Tower (D24M) workstation (Dell (China) Company Limited, Xiamen, China) was used for experimental calculation.

### 2.2. Methods

*RMSD* can be represented as the average of the structural change over the total number of atoms. *RMSD* of a protein can reveal the change in position between the conformation and the initial conformation in the simulation process. The variation trend of *RMSD* of a protein is also an important indicator to judge whether the simulation is stable or not [20]. It can be defined as:(2)$$RMSD=\sqrt{\frac{\sum_{i=1}^{N_{atoms}}(r_{i}(t_{1})-r_{i}(t_{2}){)}^{2}}{N_{atoms}}}$$
where $N_{atoms}$ is the number of atoms whose positions are being compared, and $r_{i}\left(t\right)$ is the position of atom *i* at time *t*.

*RMSF* is the average of atomic position relative to time, which can characterize the flexibility and motion intensity of amino acids during the whole simulation process [20]. It is defined as:(3)$$RMSF=\sqrt{\frac{\sum (r_{t}-r_{ref}{)}^{2}}{T}}$$
where $r_{t}$ is the conformation at a given moment, $r_{ref}$ is the reference conformation, and *T* is the total time.

*Rg* can be used to characterize the compactness of protein structure. It also shows the variation of the looseness of peptide chain in the simulation process [12,13]. It is defined as:(4)$$\begin{array}{l} Rg=\sqrt{\frac{\sum_{i}m_{i}{\left(R_{i}-R_{center}\right)}^{2}}{\sum_{i}mi}} \end{array}$$
where $R_{i}$ is the coordinates of an atom *i* and $R_{center}$ is the coordinate of the center of mass, *N* is the number of atoms.

*SASA* represents the contact area between protein and solvent or other molecules, which provides information for the contact area between protein and solvent and other molecules [20,29]. *SASA* is defined as.
(5)$$SASA=\sum (R/\sqrt{R^{2}-Z_{i}^{2}}\times D\times L_{i})$$
where *R* is the radius of an atom, $L_{i}$ is the length of the arc drawn on a selected section *i*, $Z_{i}$ is the perpendicular distance of section *i* from the center of the sphere.

A hydrogen bond is formed between the Carboxyl group and the amide group on the protein skeleton. The stability of the secondary structure of proteins is mainly maintained by the force of hydrogen bonds [12,14,24]. In molecular dynamics simulation, the number of hydrogen bonds is an important reference for the analysis of secondary structure conversion in proteins [30]. The criteria for hydrogen bonding are that the distance between the donor and the acceptor is 0.35 nm and the angle between the acceptor, donor and hydrogen atom is 30° [31]. The g_h bond program of GROMACS software (v.5.1.2, Royal Institute of Technology and Uppsala University, Stockholm and uppsala, Sweden) was applied to obtain the number of hydrogen bonds on the GLP-2 backbone.

In this study, the visualization software VMD was utilized to draw structural diagrams of the GLP-2 protein. In order to analyze the trend change in GLP-2 secondary structure under the electric field condition, the STRIDE software (v.1.9.3, University of Illinois, Chicago, IL, USA) of VMD was utilized to calculate the time evolution of the secondary structure.

## 3. Results and Discussion

### 3.1. Root Mean Square Deviation (RMSD)

The variation in *RMSD* of carbon atoms with simulation time was shown in Figure 1.

Under the condition of *E* = 0 V/nm, the *RMSD* value of GLP-2 increased from 0.4 nm to 0.57 nm and finally stabilized at 0.57 nm, with a fluctuation range of 0.4–0.6 nm. Then, electric fields were applied in the *z*-direction. It was found that under the condition of *E* = 0.2 V/nm, *RMSD* reached equilibrium quickly, and its equilibrium value was stable around 0.56 nm, with a fluctuation range of 0.3–0.7 nm. Compared with *E* = 0 V/nm, the fluctuation range of *RMSD* value under *E* = 0.2 V/nm was significantly larger in the whole simulation process. When *E* = 0.4 V/nm, the *RMSD* value increased significantly, the equilibrium value was stable at 1.15 nm, and the fluctuation range was large, ranging from 0.6 nm–1.2 nm. Under the condition of *E* = 0.5 V/nm, the stabilized *RMSD* value was around 1.25 nm and the values fluctuated from 0.8 nm to 1.3 nm. However, when *E* = 0.6 V/nm, *E* = 0.8 V/nm and *E* = 1 V/nm, the *RMSD* value was close to that of *E* = 0 V/nm. When *E* = 1 V/nm, It can be found that the *RMSD* value quickly stabilized at about 0.4 nm, and the fluctuation was very small. It was significant that the *RMSD* value changes nonmonotonically with electric filed. When 0 V/nm < *E* ≤ 0.5 V/nm, the *RMSD* value increased with the electric field intensity increasing. The conformation of GLP-2 was obviously changed as *E* increases. When 0.5 V/nm < *E* ≤ 1 V/nm, the *RMSD* value was almost the same as the case without electric field. It can be concluded that the structure is not correlated with the strength of electric field during 0.5 V/nm < *E* ≤ 1 V/nm.

### 3.2. Root Mean Square Fluctuation (RMSF)

By observing *RMSF* for each residue carbon atom of the GLP-2, it is very convenient to find out which residues play key roles in the conformational change. It indicates that the residue has relatively large flexibility if the *RMSF* of this residue has larger fluctuation in simulation [32,33].

As shown in Figure 2a, the *RMSF* values of most amino acids in the *E* = 0.2 V/nm, *E* = 0.4 V/nm and *E* = 0.5 V/nm systems were greater than that in the *E* = 0 V/nm system. It indicated that the electric field had a certain effect on the residues of GLP-2, and it consequently influenced the structure of the GLP-2 protein. It was investigated that the *RMSF* values of residues 1–17 and 20 under *E* = 0.2 V/nm were higher than in the absence of an electric field. As the electric field intensity was 0.4 V/nm, the *RMSF* values of residues 1–21 and 26–33 were higher than that under *E* = 0 V/nm. Compared with *E* = 0.2 V/nm, When *E* = 0.5 V/nm, the *RMSF* values of residues 1–6, 12–24 and 29–33 were higher than without an electric field. With electric field increasing, from 0.2 V/nm to 0.5 V/nm, the range of residue fluctuation was enlarged. The *RMSF* values of each residue in the three systems of *E* = 0.6 V/nm, *E* = 0.8 V/nm and *E* = 1 V/nm were shown in Figure 2b. The curves are almost the same as that under *E* = 0 V/nm. Therefore, it can be concluded that *E* = 0.5 V/nm was a dividing point for the conformational change in GLP-2. In the range of 0 V/nm < *E* ≤ 0.5 V/nm, the range of residue fluctuation and the *RMSF* values of some residues increased with an increase in the electric field. The electric field mainly affected residues near the *N*-terminal and the middle part. However, it did not significantly affect residues near the *C*-terminal. In the range of 0.5 V/nm < *E* ≤ 1 V/nm, the change in *RMSF* value was not obvious with the increase in electric field intensity. It was indicated that the conformational change in GLP-2 did not show a linear relationship with the electric field intensity.

### 3.3. Secondary Structure Analysis

The secondary structure of protein mainly includes *β*-folding, *β*-corner, *α*-helix, Turn and random Coil, etc. [29]. Random coil structures at *N*-terminal and *C*-terminal are existed in GLP-2, α -helix structures are found in the middle of GLP-2 structure. In Figure 3, the vertical axis was the residue sequence, and the horizontal axis was the simulation time. The secondary structure was recorded for each 0.02 ns. Thus, there were 5000 conformations during 100 ns.

From Figure 3a, when electric field was absent, the α-helix structure’s 6–26 residues remained stable. Under the condition of *E* = 0.2 V/nm, the α-helix structure between residues 5–10 was unwound at the end of simulation. According to Figure 2, the *RMSF* values of residues 1–10 were greater than 0.5 nm. We speculated that in the system of *E* = 0.2 V/nm, among the residues fluctuated by electric field, only when the *RMSF* value reached more than 0.5 nm, the α-helix structure would change. Under the condition of *E* = 0.4 V/nm, the α -helix of residues 5–13 were unwound during the simulation process. At the same time, the α-helical structure between residues 14–17 was transformed into an unstable Turn structure under the action of the electric field. The α-helix structure between residues 28–30 was transformed into a Coil structure. It can also be found in Figure 2 that the *RMSF* values of residues 5–13 were greater than 0.4 nm, which is smaller than that under *E* = 0.2 V/nm. It indicated that the increase in electric field intensity made the α -helix structure between residues 5–13 easier to unwind. In addition, the helix structure between 14–17 and 28–30 was also unwound. It means the range of residues affected by the electric field became larger. Under the condition of *E* = 0.5 V/nm, the α -helix structure between residues 16–18 was converted to a Coil structure, and the structure between residues 19–24 was unstable in the simulation process. The α-helix between residues 25–31 was transformed into a Turn structure.

When the electric field intensities were *E* = 0.6 V/nm, *E* = 0.8 V/nm and *E* = 1 V/nm, respectively, only the structure of *N* and *C* ends changed, while the α-helix structure in the middle did not change. Compared with *E* = 0.5 V/nm, the range of unwinding did not continue to increase with electric field intensity increasing, and the structure between intermediate residues kept stable. The results showed that the range of changed GLP-2 structure did not gradually increase as the electric field increases. When the electric field intensity exceeded a certain range, the α -helix structure of GLP-2 remained stable.

VMD was used to generate protein snapshots of different systems at 0 ns, 20 ns, 40 ns, 60 ns, 80 ns and 100 ns, respectively. As shown in Figure 4, it revealed the conformational changes in proteins over time. Under the conditions of *E* = 0.2 V/nm, *E* = 0.4 V/nm and *E* = 0.5 V/nm, not only were the two ends of GLP-2 transformed, but the α -helix structure in the middle part of GLP-2 was also unwound. Moreover, under the conditions of *E* = 0.4 V/nm and *E* = 0.5 V/nm, the two ends of GLP-2 were bent toward the middle, thus reducing the volume of the protein. Under the conditions of *E* = 0 V/ nm, *E* = 0.6 V/nm, *E*= 0.8 V/nm and *E* = 1 V/ nm, the structure at both ends of GLP-2 changed and the middle structure remained stable during the simulation process of 100 ns.

For comparison, the secondary structures of the GLP-2 under different external electric fields ranging from 0 to 1 V/nm are shown in Figure 5a–g. In the secondary structure of GLP-2, the purple, cyan, and white represent the α-helix, Turn, and random Coil secondary structure, respectively. It can be seen from Figure 5 that the purple helical section (residues 6−27) was disrupted in the action of 0.2 V/nm, 0.4 V/nm and 0.5 V/nm (Figure 5b–d) compared to that in the absence of an external electric field (Figure 2a). However, there is no dramatic change in the secondary structure of this section under *E* = 0.6 V/nm, 0.8 V/nm and 1 V/nm (Figure 5e–g), which is similar to the situation without an external electric field.

### 3.4. Radius of Gyration (Rg)

*Rg* can reflect the volume and shape of protein. The larger the *Rg* value is, the looser the protein is [18,34].

According to Figure 6, it can be found that the process of *Rg* obviously changed with the simulation time under each system. From Figure 6a, when *E* = 0.2 V/nm, as the stable helical structure between residues 5–10 was transformed into an unstable Turn structure, resulting in the structure of the GLP-2 protein changing from compact to loose. Thus, the volume became larger, and *Rg* became larger too. According to Figure 6b,c, the value of *Rg* under the conditions of *E* = 0.4 V/nm and *E* = 0.5 V/nm were significantly lower than that of *E* = 0 V/nm. From the corresponding secondary structure, which was shown in Figure 3c,d, it could be seen that the helix structure of the middle part of GLP-2 was unwound, and the part of the middle were bent, resulting in the decrease in protein volume. Thus, *Rg* decreased during the simulation. The *Rg* values in Figure 6d–f showed little difference compared with the *E* = 0 V/nm system. According to the analysis of the secondary structure, the secondary structure of GLP-2 protein did not change significantly, and the intermediate structure remained stable, which had little influence on the volume of GLP-2 protein. Therefore, the *Rg* values did not change significantly.

### 3.5. Solvent Accessible Surface Area (SASA)

*SASA* analyzes the contact area between protein and solvent or other molecules [35]. It is well known that when protein is affected by external factors such as temperature, electric field or chemistry, the change in structure will lead to the change in protein surface properties.

According to Figure 7a, under *E* = 0.2 V/nm, the surface area of GLP-2 increased during the simulation. The reason is the α -helix structure between residues 5–10 was unwound under the action of electric field, which led to the increasement of *SASA*. According to Figure 7b,c, the solvent accessible surface area of GLP-2 decreased during the simulation process. The intermediate structure of GLP-2 was unwound, but the structures at both ends were bent to the middle, which resulted in the reduction in the volume and the contact area between protein and solvent. As shown in Figure 7d–f, under the *E* = 0.6 V/nm, *E* = 0.8 V/nm and *E* = 1 V/nm, the solvent accessible surface area of GLP-2 did not change much compared to that without an electric field.

### 3.6. Hydrogen Bonds Analysis

Hydrogen bonds can maintain the stability of the protein structure. More hydrogen bonds represent that the protein has a more stable structure [36]. The change in the secondary structure will inevitably lead to the change in hydrogen bond content, so the degree of conformation change determines the number of hydrogen bonds [29,30,31].

Figure 8 shows the time evolution of hydrogen bonds. Here, the average value of hydrogen bonds for each 10 ns was calculated under different electric field *E*. When an electric field of 0.2 V/nm was added in the *z*-axis direction, hydrogen bonds in the simulated system began to decrease after 20 ns, and the number of hydrogen bonds reached a stable value of about 17 hydrogen bonds after 50 ns. As the electric field intensity was 0.4 V/nm, the hydrogen bond of GLP-2 had been destroyed before 10 ns, and finally the number of hydrogen bonds was stable at about 13. However, when the electric field intensity was 0.5 V/nm, the number of hydrogen bonds decreased to the lowest before 40 ns, and then increased gradually. It indicated that there was a process of recovery of GLP-2 structure under the condition of *E* = 0.5 V/nm. Under the conditions of *E* = 0.6 V/nm, *E* = 0.8 V/nm and *E* = 1 V/nm, the number of hydrogen bonds of GLP-2 changed little in the simulation process. According to the analysis of the number of hydrogen bonds in each system, it was also found that *E* = 0.5 V/nm was a turning point. When 0 V/nm < *E* < 0.5 V/nm, the number of hydrogen bonds of GLP-2 decreased as the electric field increases. For *E* = 0.5 V/nm, the number of hydrogen bonds in GLP-2 reduced and then increased in the simulation. When 0.5 V/nm < *E* ≤ 1 V/nm, the number of hydrogen bonds of GLP-2 was almost the same as that without an electric field.

### 3.7. The Charge and Polarity for Each Amino Acid of GLP-2

As discussed above, the hydrogen bonds can be easily destroyed by the influence of the electric field when *E* < 0.5 V/nm. Table 1 is the charge and polarity of each residue of GLP-2. It can be seen that residue 1 has +1 charge, residue 3 has –1 charge, and they can produce a dipole moment. Residue 20 and 21 can also produce a dipole moment. Therefore, when the electric field is not very large, such as 0.4 V/nm or 0.5 V/nm, the dipole moments of residue 1–3 and residue 20–21 align with the direction of the electric field quickly, leading to the hydrogen bonds between the dipole moments being broken (residues 3–18). The entropy of the chain increases. The chain will spontaneously fold into a conformation that tends to have hydrophobic residues inside the chain. It has also been proved in the result of *SASA* (Figure 7). Since most residues of 3–18 are hydrophobic residues, the folding of these hydrophobic residues cause the value of *SASA* to be smaller. However, when *E* reaches a certain value (*E* > 0.5 V/nm), the helix structures in the middle of the structure will recover. The reason for this may be that the electric force of the charged residues reaches equilibrium along the *z*-direction under a strong electric field. As can be seen in the snapshots in Figure 4, the helix is arranged along the *z*-direction. Considering the confinement of moving along in another direction, the residue is less free. Thus, entropy decreases and enthalpy increases, which enhance the interaction of adjacent residues. It is of benefit to recover hydrogen bonds in the middle region of the protein. Therefore, the helix in the middle region is stable, while the electric field is large (*E* > 0.5 V/nm).

## 4. Conclusions

GLP-2 is produced and secreted by the endocrine cells of the small and large intestine, and GLP-2 has many effects, such as promoting the growth and development of intestinal mucosa, proliferation of intestinal epithelial crypt cells, inhibiting the apoptosis of intestinal epithelial cells and crypt cells, and promoting repair after small intestinal injury. It is well known that there is a weak electric field inside the human body. In this paper, the influence of different electric fields on the structure of GLP-2 was discussed.

We set seven simulation systems with *E* = 0 V/nm, *E* = 0.2 V/nm, *E* = 0.4 V/nm, *E* = 0.6 V/nm, *E* = 0.8 V/nm and *E* = 1 V/nm to obtain the kinetic information of GLP-2 under different electric fields. The structural changes in each system under an electric field were analyzed by using *RMSD*, *RMSF*, *Rg*, *SASA*, secondary structure and hydrogen bond parameters. It was found that the electric field had an effect on the conformational change in GLP-2. Within a certain range of electric field strength, hydrogen bonds could be broken as the α-helix structure was disrupted. However, the degree of damage was not linear with the electric field intensity. When the electric field intensity was less than 0.5 V/nm, with the increase in the electric field intensity, the range of damaged structure became larger, and the damaged structure moved downward from the *N*-terminal. When the electric field intensity was greater than 0.5 V/nm, the conformational change in GLP-2 was not apparently affected.

The results confirmed that an electric field had a certain effect on the structure of GLP-2, which might help further exploration of the most suitable conditions for the treatment of GLP-2 related diseases. It is possible to find a physiological environment that is suitable or even possible to enhance its biological activity. In the treatment of GLP-2-involved diseases, the reduction in its physiological activity by electric fields can be avoided. These investigations provided a scientific basis for studying the physiological, pharmacological, therapeutic effects and receptor signal transduction mechanism of GLP-2.

## Figures and Tables

**Figure 1 polymers-14-02722-f001:**
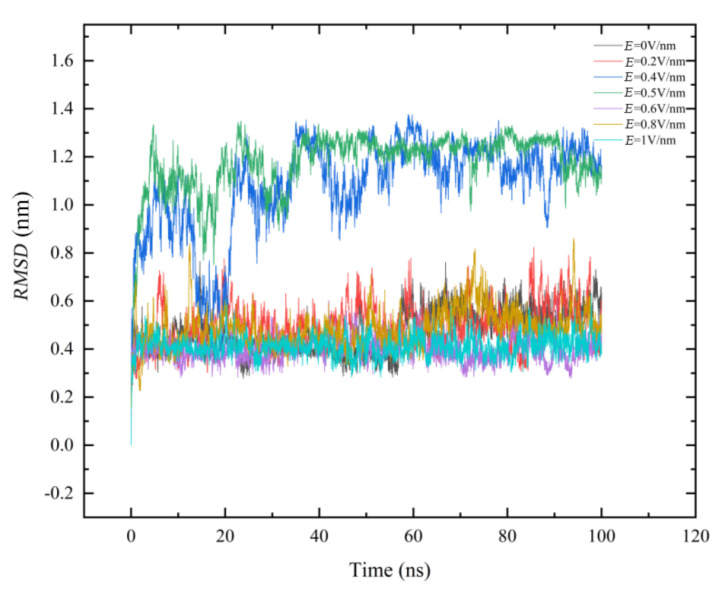
(Color online) Time evolution of backbone root mean square deviations (*RMSD*) as a function of time in 100 ns. Here, *E* = 0 V/nm (black line), *E* = 0.2 V/nm (red line), *E* = 0.4 V/nm (blue line), *E* = 0.5 V/nm (green line), *E* = 0.6 V/nm (purple line), *E* = 0.8 V/nm (yellow line), *E* = 1 V/nm (cyan line).

**Figure 2 polymers-14-02722-f002:**
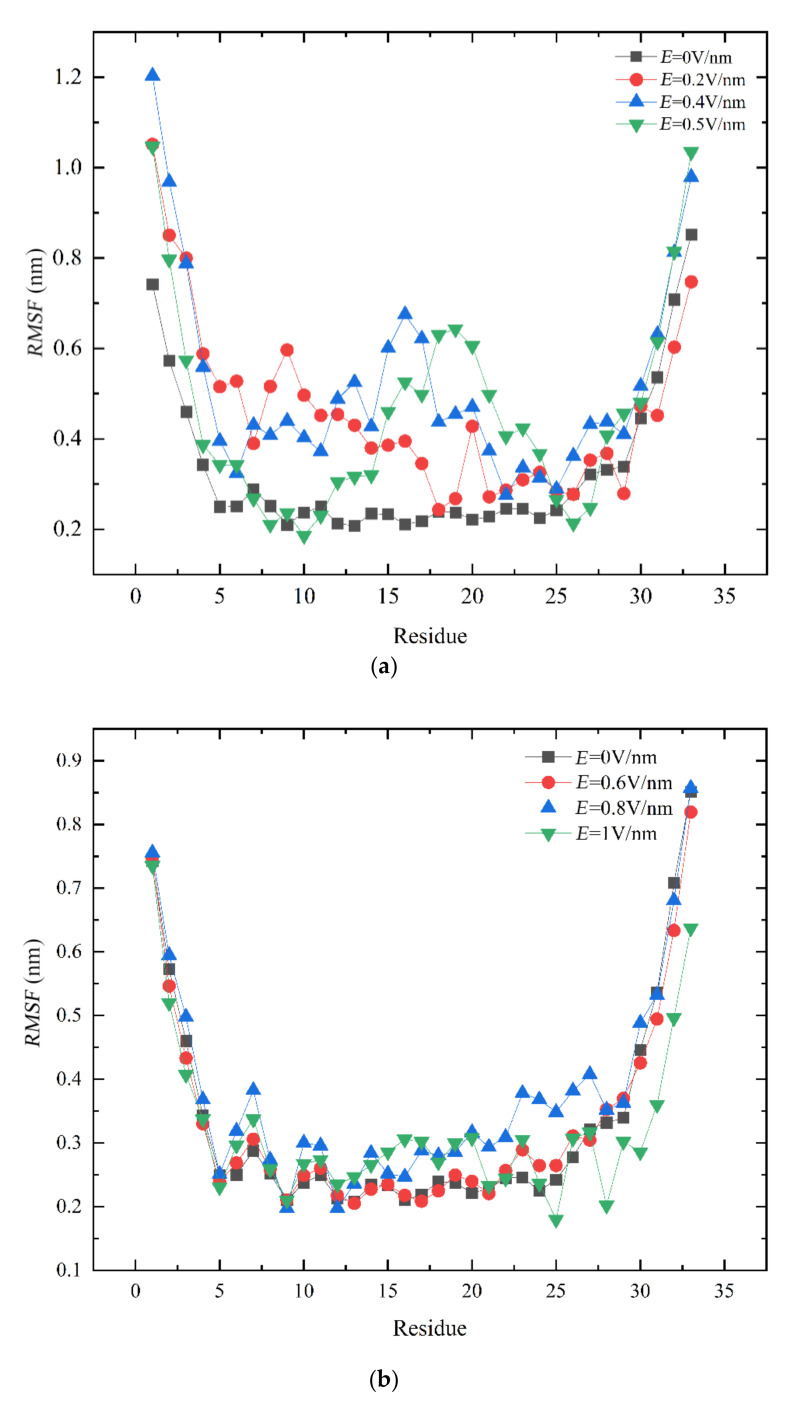
(Color online) Root mean square fluctuations (*RMSF*) of α-atoms of the protein GLP-2 in the 100 ns. (**a**), *E* = 0 V/nm (black line), *E* = 0.2 V/nm (red line), *E* = 0.4 V/nm (blue line), *E* = 0.5 V/nm (green line). (**b**), *E* = 0 V/nm (black line), *E* = 0.6 V/nm (red line), *E* = 0.8 V/nm (blue line), *E* = 1 V/nm (green line).

**Figure 3 polymers-14-02722-f003:**
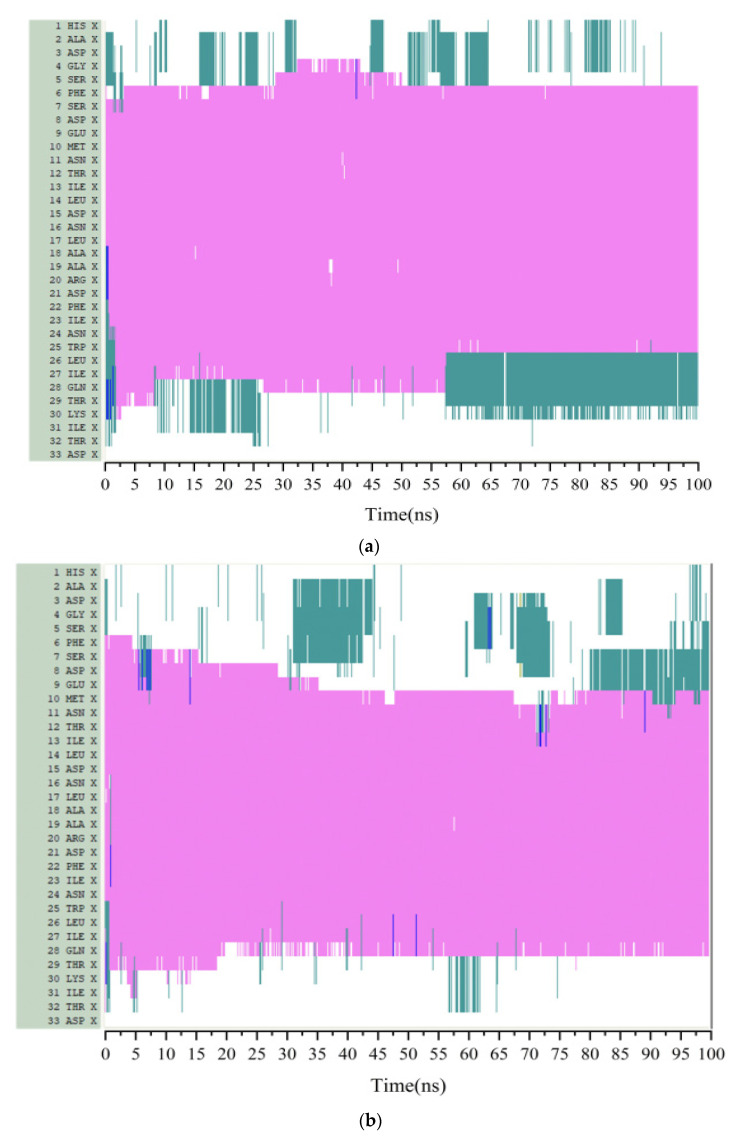
Stride evolution of secondary structures of the protein GLP-2 at different electric field strength. (**a**) *E* = 0 V/nm, (**b**) *E* = 0.2 V/nm, (**c**) *E* = 0.4 V/nm, (**d**) *E* = 0.5 V/nm, (**e**) *E* = 0.6 V/nm, (**f**) *E* = 0.8 V/nm, (**g**) *E* = 1 V/nm. (Color code: magenta color denotes α-helix, red denotes π-helix, cyan denotes turn, blue denotes 3–10 helix and white denotes coil).

**Figure 4 polymers-14-02722-f004:**
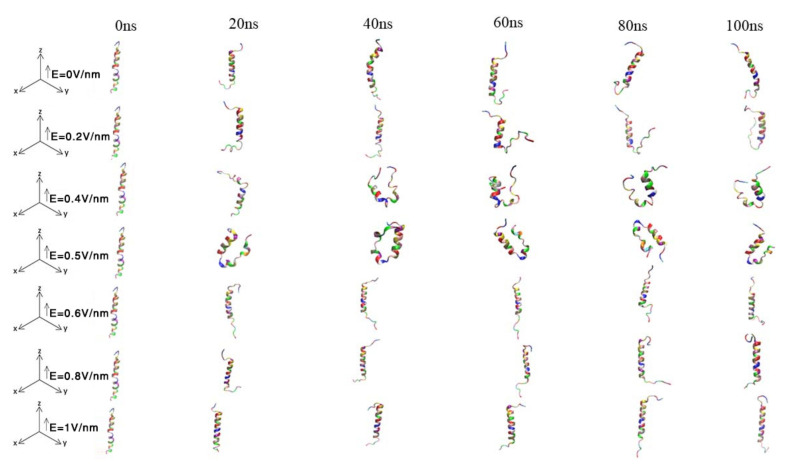
Snapshots of GLP-2 structure at 0 ns, 20 ns, 40 ns, 60 ns, 80 ns and 100 ns for different systems. Different colors correspond to different residues (available in VMD).

**Figure 5 polymers-14-02722-f005:**
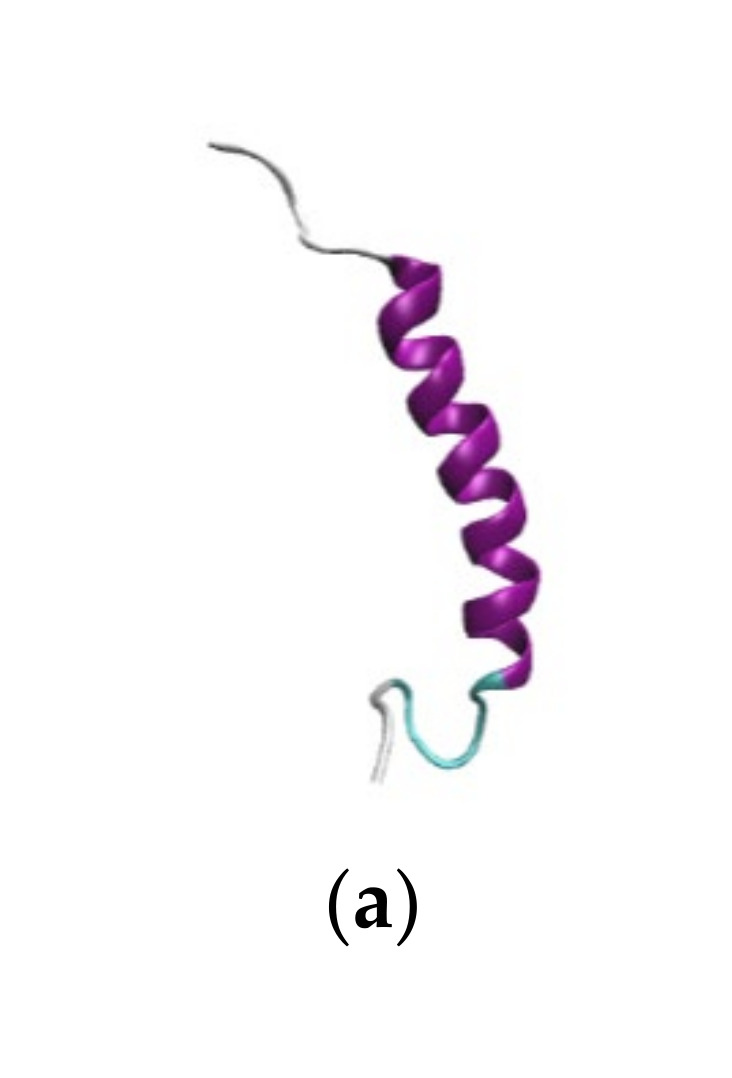
Three-dimensional structures of GLP-2 after 100 ns MD simulations. The structures without external electric field (**a**) and under the external electric field of 0.2 V/nm (**b**), 0.4 V/nm (**c**), 0.5 V/nm (**d**), 0.6 V/nm (**e**), 0.8 V/nm (**f**) and 1 V/nm (**g**).

**Figure 6 polymers-14-02722-f006:**
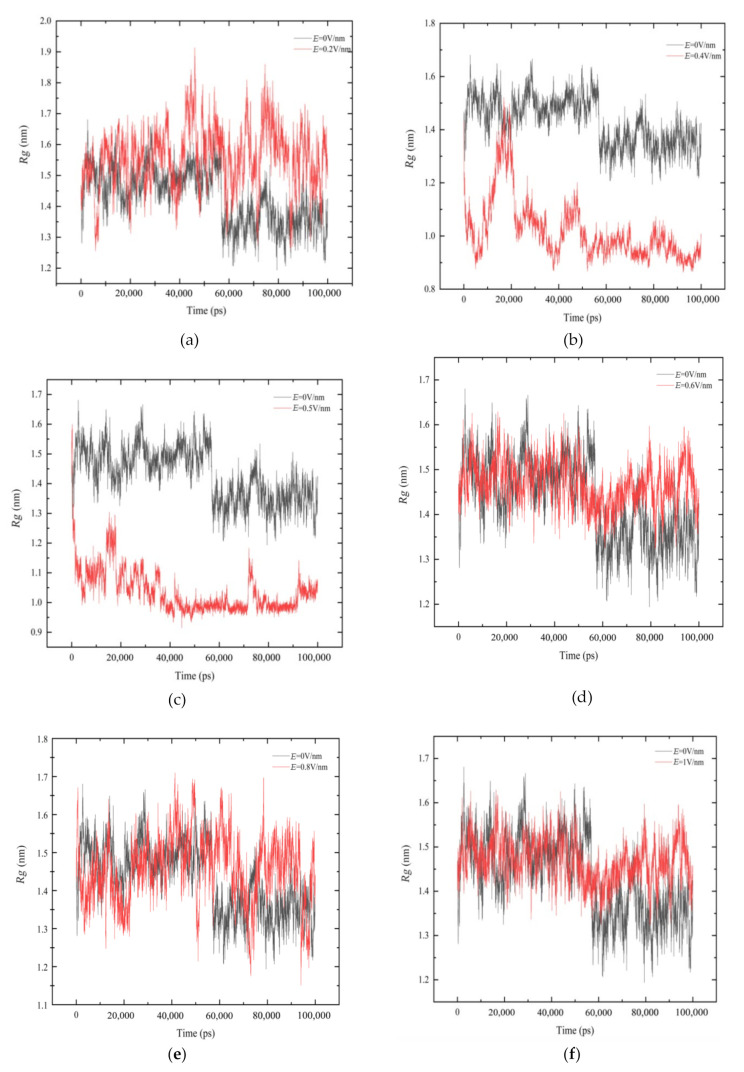
Time evolution of radius of gyration (*Rg*) of the protein GLP-2 are shown under different electric field strength (**a**) *E* = 0.2 V/nm, (**b**) *E* = 0.4 V/nm, (**c**) *E* = 0.5 V/nm, (**d**) *E* = 0.6 V/nm, (**e**) *E* = 0.8 V/nm, (**f**) *E* = 1 V/nm.

**Figure 7 polymers-14-02722-f007:**
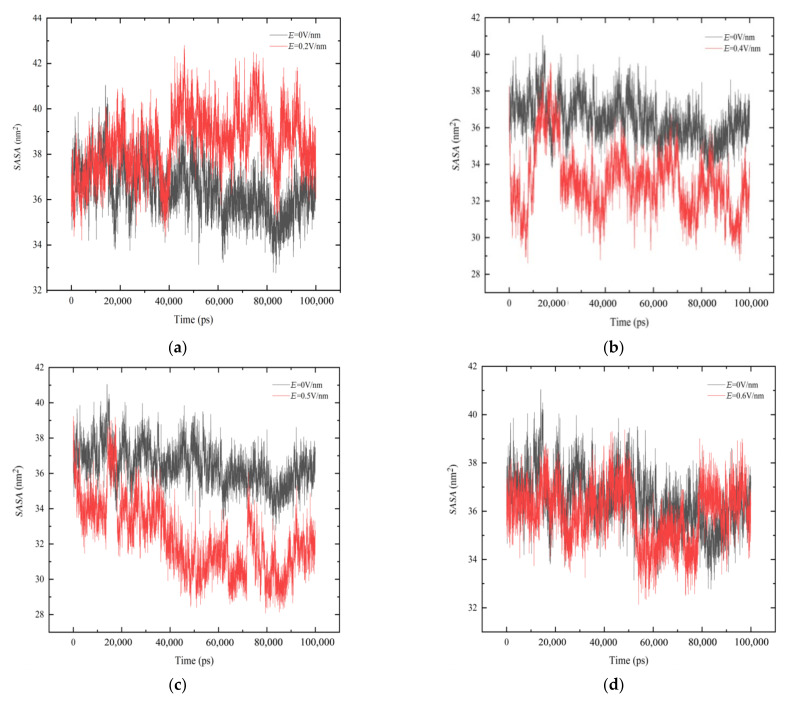
Changing in solvent accessible surface area (*SASA*) is shown as a function of time in 100 ns under different electric field strength. (**a**) *E* = 0.2 V/nm, (**b**) *E* = 0.4 V/nm, (**c**) *E* = 0.5 V/nm, (**d**) *E* = 0.6 V/nm, (**e**) *E* = 0.8 V/nm, (**f**) *E* = 1 V/nm.

**Figure 8 polymers-14-02722-f008:**
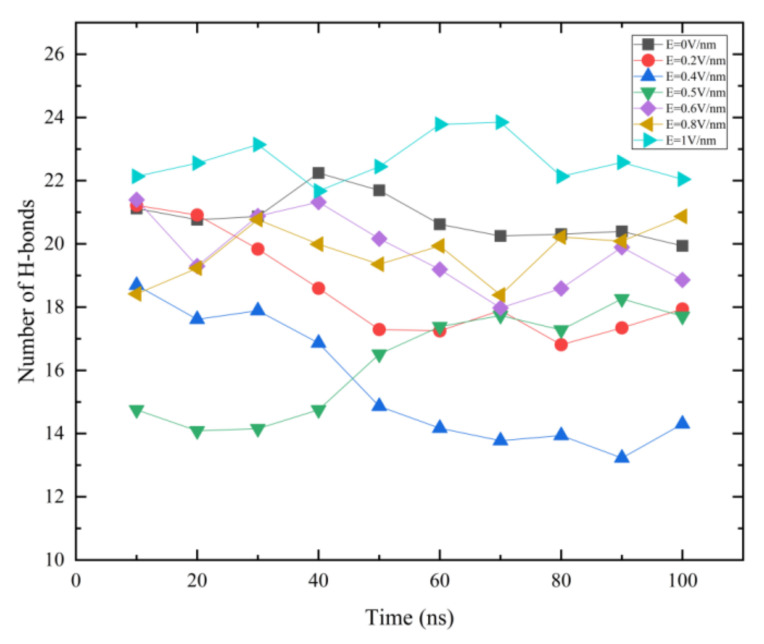
(Color online) Time evolution of hydrogen bonds are shown as a function of time in 100 ns. There are 10 points on each line due to the average of each 10 ns. The symbol coding scheme is as follows: *E* = 0 V/nm (black line), *E* = 0.2 V/nm (red line), *E* = 0.4 V/nm (blue line), *E* = 0.5 V/nm (green line), *E* = 0.6 V/nm (purple line), *E* = 0.8 V/nm (yellow line), *E* = 1 V/nm (cyan line).

**Table 1 polymers-14-02722-t001:** The Charge and Polarity for Each Amino Acid of GLP-2.

**The amino acid number**	**1**	**2**	**3**	**4**	**5**	**6**	**7**
Abbreviation of amino acid	H	A	D	G	S	F	S
Polarity	Acidic	Hydrophobic	Acidic	Hydrophobic	Hydrophobic	Hydrophobic	Hydrophobic
charge	+1.0	0.0	−1.0	0.0	0.0	0.0	0.0
**The amino acid number**	**8**	**9**	**10**	**11**	**12**	**13**	**14**
Abbreviation of amino acid	D	E	M	N	T	I	L
Polarity	Acidic	Acidic	Hydrophobic	Hydrophobic	Hydrophobic	Hydrophobic	Hydrophobic
charge	−1.0	−1.0	0.0	0.0	0.0	0.0	0.0
**The amino acid number**	**15**	**16**	**17**	**18**	**19**	**20**	**21**
Abbreviation of amino acid	D	N	L	A	A	R	D
Polarity	Acidic	Hydrophobic	Hydrophobic	Hydrophobic	Hydrophobic	Acidic	Acidic
charge	−1.0	0.0	0.0	0.0	0.0	+1.0	−1.0
**The amino acid number**	**22**	**23**	**24**	**25**	**26**	**27**	**28**
Abbreviation of amino acid	F	I	N	W	L	I	Q
Polarity	Hydrophobic	Hydrophobic	Hydrophobic	Hydrophobic	Hydrophobic	Hydrophobic	Hydrophobic
charge	0.0	0.0	0.0	0.0	0.0	0.0	0.0
**The amino acid number**	**29**	**30**	**31**	**32**	**33**		
Abbreviation of amino acid	T	K	I	T	D		
Polarity	Hydrophobic	Acidic	Hydrophobic	Hydrophobic	Acidic		
charge	0.0	+1.0	0.0	0.0	−2.0		

## Data Availability

Not applicable.

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
