# Peer review of "Conformational Dynamics of Glucagon-like Peptide-2 with Different Electric Field"

_polymers, 2022, doi:10.3390/polym14132722_

Round 1

Reviewer 1 Report

In the present work "Dynamics of Glucagon-Like Peptide-2 with different electric 2 field", Su et al. explore the effects of external electric fields on the dynamics of a test case protein. The overall idea is good especially now that literature is growing on the topic of external fields on proteins. However, I have major concerns about the structure of this work. Here, there is a list of my concerns:

* The presentation of the figures must be improved. Figures look currently too crowded. I recommend separating the curves into panels or showing only the most relevant values for the electric field. Another idea is to use 3D plots (see for instance: https://link.springer.com/article/10.1007/s10858-020-00343-9) Otherwise, one cannot see any relevant features.

* line 153: which residue is more flexible?

* explain all equations in detail. What is R in all equations? The root mean square deviation is not well explained, in the form it is written one cannot compute the RMSD. Please check carefully the Methods section.

* in general, the residues close to the end are more flexible than the ones in the middle. How much the current results can be modified if those residues are not considered when computing the RMSD, RMSF, and gyration radius? 

* do you have a deeper understanding of why E=0.5V/nm is a turning point? BTW, inflection point concept would involve a second derivative.

* An arrow showing the field direction would be useful in Figure 4

* I am curious why you chose the Gromacs version 5.12 which is old? The latest one includes the features for simulations with electric fields and it is much faster.

* 100 ns is a short simulation time these days. It is strongly recommended to extend the trajectories possibly up to 1 microsecond.

* extensive English language editing is advised. 

Minor changes:

* line 113: "carbony group" check spelling

* line 147: "as E increasing" 

* line 180: stranged symbols were introduced

Reviewer 2 Report

Paper titled (Dynamics of Glucagon-Like Peptide-2 with different electric field) by Su et al. tested the effect of electric field on GLP-2 protein and concluded that  the structural change of GLP-2 was not clearly correlated with the strength of the electric filed. This is a novel study and the aim is straightforward however, authors in general needs to further identify the aim of the study and the conclusion and future directions to enhance the utility of this study.

1- Title : needs to be informative & state what was the dynamics results

2- Introduction: while stating the aim, this part needs to further exploration of the importance of the study 

3- In the conclusion, what would be the positive  consequence upon this important study? this needs further exploration.

4- Authors should give the source of chemicals, kits and antibodies completely and consistently (code, company, town, state and country) & version for software

5-  Methods in general lacks references at many occasions.

6- Use abbreviations consistently

Reviewer 3 Report

The authors performed a computational study on the GLP-2 peptide in aqueous environments under static external electric field (EF). They found the peptide molecule respond to EF differently under different strength of the EF. The middle helix was found unwind under certain strength of EF while showing certain stability under ether lower or higher strength of EF. This structural change is also reflected from other characterizations such as RMSD, RMSF, SASA, HBond etc. This is an interesting research work, and the finding was unexpected as well. I would like to ask some questions below to verify some facts and details before making any decision:

(1)    The external EF will definitely affect the conformation of the peptide chain, and it can also affect the orientation/distribution of the solvent molecule due to the polarity rotation. If I understand correctly, the electric field is acting abruptly at very beginning of the simulation, and this action could inject a lot of energy to the whole simulation system and heat up the water molecules and break the hydrogen bond network and the secondary structure of the peptide. The author could check the temperature of the system and get to know if this effect exists or not. From this perspective, this helix unwinding effect could come from the initial high energy injection instead of helix responding to the electric field. If this is true, then the author should be very careful about how to describe the simulation results about the true reason for this unexpected result. Could the author testing on the E=0.5 V/nm sample with a slowly increasing EF instead of a step-like EF to check if the unwinding still exists? I hope this could be helpful for understanding the true reason for the helix unwinding under EF.

(2)    Some plots have shown an additional frame outside the plot like a table boundary. Could the author remove them?

(3)    Fig 3 should be replotted for more clarity and readability. It looks really bad in current form. Panels should be aligned together as well.

(4)    Language polishing is needed, some typos are found.

Round 2

Reviewer 1 Report

The manuscript improved considerably in quality w.r.t. the previous version. I still have some comments that need to be addressed before considering this work for publication:

* Line 327: "It is well known that there is a weak electric field inside the human body", could you cite literature about this point? How is the electric field produced? In what parts of the body? What are the magnitudes in V/m for instance for the experiments in Ref. 19?

* Regarding the methods section, for how long did you equilibrate the systems? Is this equilibration time taken into account in the 100ns? If that is the case, you are strongly advised to extend the simulation time. I still consider that 100ns is just a too short time for current simulations.

* Figure 2 is hardly readable as I mentioned in my previous review.

* In the calculation of RMSD, RMSF, I am missing a sentence saying that these formulas can be applied after some alignment is performed on the trajectories.

* Regarding the Gromacs version, the simulations with electric fields are supported in the latest versions and they should run faster even on CPUs.

* Line 15: "The investigation about the effect of electric field on the structure of GLP-2 16 can provide some theoretical basis for the biological function of GLP-2 in vivo" could you please provide literature or evidence that the magnitudes of the electric fields you studied in this work can be found in vivo systems specially aligned fields? In Ref. 19 (explicit magnitudes not cited only ratios), the idea seems to be measuring the response of applying an external signal. But by no means this field is aligned as in the present work and statistically, such a response would have a zero effect on any protein in vivo.

* I strongly recommend increasing the resolution of the figures.

Reviewer 2 Report

I recommend accepting the current form 

Author Response

Thank you for the reviewer's recommendation.

Reviewer 3 Report

It looks like the charge distribution of the peptide chain leads to this unique response between the chain structure and external field. I suggest authors adding additional one figure (show the charge/dipole/sequence characteristics of this peptide chain) and discussing more about the reason why the unwinding only happened at weak electric field. Try to discuss about the enthapy and entropy compitition of the peptide chain under electric field. I think it will make this paper much more interesting.

Round 3

Reviewer 1 Report

The readability of the manuscript improved considerably. I will accept it for publication as it is in its present form.